# Competitiveness of Namibia's Agri-Food Commodities: Implications for Food Security

Yonas T. Bahta [1,*] and Salomo Mbai [2]

1 Department of Agricultural Economics, University of the Free State, Bloemfontein 9300, South Africa
2 Department of Agriculture and Natural Resources Sciences, Agricultural Trade Policy Institute (ATPI), Namibia University of Science and Technology, Windhoek Private Bag 13388, Namibia
* Correspondence: bahtay@ufs.ac.za

**Abstract:** The global literature widely uses Revealed Comparative Advantage (RCA), Lafay Index (LFI), Export Diversification (EDI), Hirschman (HI), and Major Export Category (MEC) to analyze international trade flows, though agricultural trade, specifically agri-food commodities consisting of food and feed commodities, is neglected in empirical works. Furthermore, the determinants of Revealed Comparative Advantages and the Lafay Index have received little attention, with little focus on the relationships between food insecurity as measured by the Household Food Insecurity Access Scale (HFIAS) and the aforementioned factors, including the RCA and LFI indices as explanatory variables with other macroeconomic variables. The purpose of this study was to ascertain the competitiveness of Namibia's agri-food products, the factors that influence it, and their implication for food security. This study attempts to answer which factors affect agri-food comparative advantage, which agri-food commodities have a comparative advantage and disadvantage, and what the implications are to food security. The study employed regression analysis, the Household Food Insecurity Access Scale, and various indices. Revealed Comparative Advantage and Lafay Index indicated a mixed result during the study period. Export Diversification and Hirschman indices indicated a less concentrated trade structure throughout the study period. The results of the market structure of the international agri-food market estimated by the Major Export Category revealed that Namibia was unduly dependent on agri-food commodities incorporated in this study. The regression result showed a significant negative influence of labor and land productivity on the aggregated RCA and LFI for agri-food commodities. Land productivity and GDP per capita impact the degree of food insecurity in Namibia. The study concludes that Namibia was not dependent on international trade of agri-food commodities and had little bearing on food security. These analyses enlighten policymakers about the competitiveness of the agri-food business and its implications for food security through evidence-based policy development.

**Keywords:** revealed comparative advantage; Lafay Index; land productivity; labor productivity; Household Food Insecurity Access Scale; concentrated trade structure

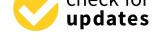



## 1. Introduction

The competitiveness of an economy and its industries determines how well a country integrates into global markets, performs in trade, and creates value in supply chains [1]. In response to global integration, market liberalization has shaped agricultural markets, which have imposed competitiveness for survival. Consequently, sustainable growth depends on the competitiveness of agriculture and its impact on food security.

The African Union recognized in the Malabo Declaration, signed in 2014 that agri-food can play a critical role in food system transformation, poverty reduction increased opportunities for women and youth in agribusiness, increased intra-African agricultural trade, and reduced food insecurity and hunger. Despite the significant opportunities for economic growth, poverty reduction, and food security presented by agri-food in Africa,

the sector's growth and modernization continue to lag. Some of the challenges include a lack of technical training and human resources, limited access to land and financing, varying access to raw materials, high operational costs, and poor infrastructure. Smaller businesses suffer from such challenges because they find it difficult to expand or produce higher-quality goods [2].

Even when faced with challenges, increased competitiveness can help agribusiness, agriculture, and agri-enterprises increase their income levels and reduce food insecurity in the long run [3]. Technical, environmental, social, economic, and political changes are among the challenges [4]. As a result, strategies, programs, and policies at the farm and industry levels are required to overcome challenges.

Agriculture is one of Namibia's most important industries, employing roughly 70% of the country's population directly or indirectly, primarily in the subsistence sector. It contributed more than 4% slightly to GDP from 2014 to 2018 and 6.6% in 2019. The Republic of Namibia's government is guided by long-term development objectives outlined in National Development Plans (NDPs) and the Vision 2030 strategy. As a result, the government's goal is to ensure agricultural productivity and food security in accordance with Vision 2030. In Namibia's fifth National Development Plan (NDP5), the agricultural sector and food security goal is to reduce the proportion of food-insecure people from 25% in 2016 to 12% and increase food production by 30% cumulatively between 2017 and 2022 [5].

Food systems encompass a wide range of activities in the production, processing, distribution, marketing, preparation, consumption, and disposal of agricultural goods, while also considering the value chain's social, ecological, and economic interactions. This includes food security as well as the broader set of food-related systems, which can consist of traditional, modern, and informal channels [6]. The Namibian food system can provide enough food through either production or trade.

Competition in domestic and international trade is essential for the globalization of economic activities. Participating in international trade is imperative to improve efficiency and global competitiveness [7]. Namibia's participation in various trade agreements also affects the agricultural sector. Taking part in international trade agreements reduced distortions caused by indirect export subsidies and transformed the structure of agri-food trade flows and comparative advantages [8].

Agricultural commodities continue to be subject to public protection for several reasons. The business faces three fundamental problems: stability, income, and foreign trade. The market for most agricultural products is more unstable than is necessary to optimize resources and manage buyers' expenditures efficiently, though it is not unanimous. According to Khor [9], the world's most pressing problems are the distorting nature of global agriculture trade, including market instability, income losses due to terms-of-trade declines, and international agricultural trade negotiations. Agricultural prices, output, and income fluctuate significantly because of inelastic demand and supply, uncertain foreign markets, weather, insects, and diseases. It is common for legislation intended to protect farmers and programs dealing with the marketing of agricultural products to use the term "stabilization". Whether this goal can be achieved through the policies in place and whether they are sufficiently integrated to create an environment that increases efficiency to enhance competitiveness remains to be seen. This study attempts to answer which factors affect the agri-food comparative advantage of Namibia, which agri-food commodities have a comparative advantage and disadvantage, and what are the implications to food security.

The Namibian government has implemented several policies and strategies to improve Namibia's agricultural sector's competitiveness in both domestic and international markets. These policies and strategies aim to fully exploit the open global market based on the country's perceived competitive advantage. Despite the policies, the lack of studies examined competitiveness considerations in Sub-Saharan Africa and Namibia in particular. Previous research on competitiveness focused on agribusiness, commodities like timber and soybeans, agri-processing, economic growth, and the food supply chain [7,10–21].

These studies, however, did not focus on factors influencing agri-food competitiveness and its implications for food security. Bahta and Willemse [10] examined competitive advantage in soybean production using a variety of metrics. They discovered that the nominal rate of protection was more significant than the effective rate of protection, implying that the tariff applied to output was more significant than the tariff applied to inputs. The structure of the tariff schedule may have a substantial impact on efficiency. Esterhuizen et al. [11] and Sharma et al. [12], using various indices, the agribusiness and agri-processing sector's competitiveness, structure, and value addition were assessed.

Esterhuizen et al. [11] found that decreasing competitiveness in the value chains suggests that industry value-adding opportunities are constrained. Sharma et al. [12] discovered that the extent of value addition in the processing sector was approximately 53%, and financial viability ratios revealed a high current ratio but a lower quick ratio (acid test) in most of the processing industries, indicating that many sectors have significant unsold inventories. Hencion and McIntyre [13] reviewed the literature, looked at the effects of the food supply chain, and discovered that the biggest influences on food supply chains were identified as consumers, then retailers. Mbai et al. [14] examined timber export competitiveness, and their research found that, due to Namibia's reliance on natural forests, timber export competitiveness is not sustainable, export patterns are highly reliant on the volumes and values of exported timber, and Namibia's timber exports are not competitive.

Louw et al. [15] identified factors impeding agri-processing development. They discovered that the small wheat-milling and baking industries have relatively high entry barriers. These barriers include the need to establish a market, learn about the wheat-milling and baking industries, maintain a well-maintained infrastructure, have the necessary marketing and management skills, and have the necessary cash flow. Mlambo et al. [16] examined the role of agricultural commodity exports in economic growth, and the findings showed that exports of processed agricultural products had a favorable impact on it. Agriculture exports that have not been processed, however, have a negative impact on economic expansion. This demonstrates how manufactured agricultural exports make a significant contribution to economic growth.

Bojnec [17] analyzes regional and global agricultural trade patterns and discovers that trade and comparative advantage indicators are distorted by trade and other policy barriers, which may artificially boost exports through export subsidies or protect domestic production using tariffs and non-tariff trade measures. The level, composition, complementarities, pattern of agri-food trade specialization, and variations in agri-food relative trade advantage/disadvantage are all examined by Bojnec and Ferto [7,18,19] with an eye toward their implications for food policy. Bojnec and Ferto [7] discovered that consumer-ready foods have lower comparative trade advantages than bulk primary raw agricultural commodities. This suggests that food processing and international food marketing have weaker competitiveness. Bojnec and Ferto [18] emphasized that trade deficits in agri-food products have increased due to trade specialization patterns in Southeastern Europe. Additionally, agri-food export markets are heavily concentrated in bulk raw commodities and lack export specialization. According to Bojnec and Ferto [19], Central European nations had a higher proportion of products with relative trade disadvantages and a greater significance of one-way trade.

Sattar et al. [20] assessed the competitiveness of the food processing cluster in Namibia using the Porter Cluster Competitiveness framework. They found that within this framework, the four determinants of competitiveness are demand conditions (markets), factor conditions, the firms' structure and strategies, and the related and supporting industries (cluster foundations). Bahta [21], on the other hand, determines the competitiveness of South African agri-food products as well as the factors that influence them and revealed a mixed outcome for RCA and LFI of agricultural and food commodities from 2000 to 2018. Regression analysis of the variables affecting the competitiveness of agri-food commodities produced various results. GDP per capita and agricultural productivity both had a positive effect on comparative advantage. Agri-food commodities had both positive and negative

impacts on macroeconomic stability. South Africa did not rely on international trade from the agri-food sector and had a less concentrated trade structure.

This study aimed to determine the level of competitiveness of Namibian agri-food products, identify the factors that influenced agri-food competitiveness, and its implications for food security using various indicators and regression analysis. Notably, there are no studies examining food security implications. The article adds to the body of knowledge by explaining the comparative trade benefits of Namibia's agri-food product markets. The findings may have a more comprehensive application for those who engage in agri-food trading daily. The empirical findings may also aid in evaluating policy implications for competitive agri-food trade and its effects on food security for strategy and policymakers in the agri-food sector. The non-participating nations in multilateral trade negotiations and outside of preferential trading arrangements are more likely to lose [22]. The next section provides the data and methods, followed by results, including policy implications discussed in Section 4, while Section 5 provides the conclusions.

## 2. Materials and Methods

### 2.1. Description of the Method, Data, and Sources

Various models and indicators were used to understand better the trade structure, trade pattern, and agri-food competitiveness. These included the theoretical and empirical concepts of the RCA, LFI, EDI, HI, MEC indices, and others. Furthermore, from 2000 to 2021, a model for identifying the drivers that give Namibian agri-food a competitive advantage was estimated using RCA and LFI as outcome variables against explanatory variables based on various data sources. HFIAS was used as an outcome variable against explanatory variables, including RCA and LFI applied to look at the implications for food security.

Both primary and secondary data were used in this study. The secondary statistics from the commodity trade data from the United Nations were used to calculate RCA, LFI, EDI, HI, and MEC indices [23]. The Food and Agriculture Organization provided the value of agricultural production in US dollars at constant prices (land and labor productivity) [24]. The World Bank provided data on macroeconomic stability, inflation (INF), and consumer price growth (annual %), as well as gross domestic product per capita in current US dollars (GDPpc) [25]. Household Food Insecurity Access Scale (HFIAS) was obtained using primary and secondary data from various sources.

Based on the Standard International Trade Classification Revision 4, the agri-food goods and associated Division codes evaluated in this study (2000 to 2021) [26]. Specifically, (i) Food and live animals: Live animals (00); Meat and meat preparations (01); Dairy products and birds' eggs (02); Fish, crustaceans, mollusks, and aquatic invertebrates and preparations thereof (03); Cereals and cereal preparations (04); Vegetables and fruit (05); Sugar, sugar preparations, and honey (06); Coffee, tea, cocoa, spices, and manufactures thereof (07); Miscellaneous edible products and preparations (09), (ii) Tobacco (tobacco and manufactured tobacco substitutes) (12), (iii) Crude materials, inedible (raw hides and skins (other than fur skins) (21), and leather; silk (26) and (iv) Animal vegetable oils, fats, and waxes—Animal, vegetable fats and oils, cleavage products, etc. (43).

### 2.2. Revealed Comparative Advantage and Lafay Index

The relative trade performance of Namibian agri-food commodities is the subject of the RCA theory and model. In this study, the RCA was used to assess the competitiveness of agri-food in Namibia. Because of the distortion it introduces, this index has been criticized in empirical trade analysis [27–29]. Lafay [30] proposes to address these shortcomings by developing weighted indicators of trade balance contribution that, while containing interesting information, are ambiguous in measuring trade specialization. Because of the increasing importance of intra-industry trade in agri-food trade due to the integration process, economic growth, and macroeconomic fluctuations during the studied period.

The RCA index continues to be the most widely used tool in empirical trade analysis despite a number of significant drawbacks, including the distortion it introduces, the asymmetric value problem, and the problem with logarithmic transformation [31].

According to Balassa [32,33], the model to compute the Balassa RCA index is (Equation (1)):

$$\mathrm{RCA_i} = \frac{\dfrac{X_{ij}}{\sum\limits_{i} X_{ij}}}{\dfrac{\sum\limits_{j} X_{ij}}{\sum\limits_{i}\sum\limits_{j} X_{ij}}} \tag{1}$$

where: $X$ export in US$

$X_{ij}$ exports of the sector "$i$" of country "$j$"

$\sum\limits_{i} X_{ij}$ Total exports of the country "$j$"

$\sum\limits_{j} X_{ij}$ World exports of the sector "$i$"

$\sum\limits_{i}\sum\limits_{j} X_{ij}$ Total "world" export

When RCA > 1, Namibia is said to be specialized in agri-food commodities or Namibia has a comparative advantage compared to reference countries. It means that Namibia had a competitive advantage in agri-food products. Namibia faced a comparative disadvantage when RCA < 1, Namibia is said not to be specialized (under-specialized) in agri-food commodities, or Namibia has a comparative disadvantage compared to reference countries [28]. The RCA was applied in several sectors, commodities, and national studies [10,13,28,34–36]. According to Istudor et al. [29], interpreting the Balassa Index can be difficult and may not explicitly assist in defining the degree of competitiveness and its dimensions. For example, a country may experience a decrease in competitiveness while maintaining a specific product or service advantage; however, it may also be competitive without a comparative advantage. They also pointed out that one limitation of the Balassa Index is that it focuses solely on trade export performance, ignoring the factors contributing to competitiveness. Furthermore, Esterhuizen et al. [37] argued that the difficulty quantifying competitiveness had led Balassa to focus on trade patterns rather than underlying resources, subsidies, and prices. Balogh and Jámbor [38] acknowledged that the Balassa Index neglects the effects of agricultural policy and, under certain circumstances, can exhibit asymmetric values.

There have been numerous attempts to overcome the shortcomings of RCA. Alternatives include export-only indices (such as the EDI, HI, and MEC indices, which include only export variables (RCA), as well as trade-cum production indices (LFI), which include both trade and production variables [39].

Theoretically, robust measures for estimating comparative advantage would be trade-cum-production indices like LFI. According to Bowen [40], the RCA index is partly a "failure of the theoretical framework" because it separates exports and imports when comparative advantage is a net trade concept. Bowen suggests an alternative index that includes production variables. Additionally, Lafay [30] notes that the RCA method eliminates the influence of macroeconomic variables; the same rationale for which Balassa excluded import variables and only included export variables can be applied to import variables. That is, while tariffs and other protective measures on the import side have been accused of causing bias in trade performance measures through successive multilateral negotiations, the same argument can be made on the export side, where subsidies or voluntary export restraint have been increased.

While trade-cumulative-production indices have been more focused on better connecting with theory, exports-only indices have been more focused on transforming and adjusting the existing RCA index to overcome its disadvantages, particularly its asymmetric property, while maintaining its practicality, ease of use, and simplicity [41].

The Lafay Index (LFI) [30] was also used as an alternative measure of comparative advantage, as expressed in Equation (2):

$$LFI_j^i = 100 \left( \frac{x_j^i - m_j^i}{x_j^i + m_j^i} - \frac{\sum_{j=1}^n (x_j^i - m_j^i)}{\sum_{j=1}^n (x_j^i + m_j^i)} \right) \frac{x_j^i + m_j^i}{\sum_{j=1}^n (x_j^i + m_j^i)} \tag{2}$$

where: $x_j^i$ exports of the sector "$i$" of country "$j$"

$m_j^i$ imports of the sector "$i$" of country "$j$"

$n$ number of items (agri-food product)

The LFI provides a more thorough examination of Namibia's agri-food commodities export participation. The normalization of each product or sector is obtained by weighting each product's contribution regarding the respective importance in the agri-food trade. Because the LFI measures each product's contribution to the overall normalized agri-food trade balance, the following relation holds $\sum_{j=1}^N LF_j^i = 0$.

Namibia demonstrated comparative advantages when LFI > 0, and the higher the value of LFI, the higher the level of specialization in the agri-food trade or LFI > 0 holds for a certain product $j$, then trade specialization is revealed; the larger value indicates a higher degree of the product's trade specialization. Similarly, negative values imply trade de-specialization.

*2.3. Export Diversification Index/Concentration Index*

The Export Diversification Index (EDI) was used to measure the agri-food sector's export performance and competitiveness in Namibia, as indicated in Equation (3):

$$EDI_j = \frac{\sum_i |h_{ij} - h_i|}{2} \tag{3}$$

where: $h_{ij}$—share of the commodity "$i$" in the total exports of country "$j$".

$h_i$—share of the commodity in world exports.

The Hirschman index (HI) was used in this article to estimate the market structure of the international agri-food market [34]. An analogous export diversification metric, the Hirschman index (HI), was used by the United Nations Conference on Trade and Development (UNCTAD) [42]. Equation (4) shows the HI as the shares of Namibia's agri-food commodities:

$$HI_j = \sqrt{\left( \sum \left( \frac{x_i}{X} \right)^2 \right)} \tag{4}$$

where: $X_i$—export of all commodities "$i$"

$X$ export of commodity "$i$"

The World Bank [43] highlighted that the EDI and HI indices range from 0 to 1. The lower value, the less concentrated Namibia agri-food exports. Thus, a value close to zero indicates that Namibia has a less concentrated agri-food trade structure.

*2.4. Major Export Category*

The major export category (MEC) classifies agri-food commodities that account for 50% or more of total agri-food exports and account for the majority of Namibia's j exports. A share of total country j exports is computed and rated for each exporting agri-food product i (Equation (5)). Namibia's exports are said to rely heavily on a single agri-food commodity category.

$$MEC_i = \frac{X_{ij}}{\sum_{i=1}^n X_{ij}} \times 100 \tag{5}$$

where: $X_{ij}$ exports of the sector "$i$" of country "$j$"

$\sum_{i-1}^n X_{ij}$ Total exports of the country "$j$"

The economy is classed as diversified if no single agri-food commodity accounts for 50% or more of total exports.

### 2.5. Land productivity (LAND) and Labor Productivity (LABOUR)

Porter's diamond model [44] represents a framework for investigating why certain economic sectors, such as agri-food, within a country, are competitive internationally. Furthermore, factor conditions are one of the data requirements that is displayed in the form of a diamond. Natural capital and other types of resources, such as financial, technological, labor, and others, are examples of factor conditions. Land and labor productivity were included in this study because these two variables are appropriate for assessing competitiveness levels. This factor condition diamond attribute highlights how various endowments contribute to international competitiveness. In this study, with the goal of providing a multidimensional analysis of Namibia's agri-food sector's competitiveness was assessed using Revealed Comparative Advantage (RCA). As a result, Land productivity is calculated as the value of agricultural production in USD (constant 2005 USD prices) per hectare of agricultural land. Labor productivity is calculated as the value of agricultural production in USD (constant prices) per number employed in agriculture.

### 2.6. Food Security Implications/HFIAS

The link between agricultural and agri-food competitiveness, international trade flows, and food security measured by HFIAS has long been under consideration by policymakers and scholars [45,46]. From a scientific standpoint, this link becomes more compelling, particularly in light of the COVID-19 pandemic, the energy crisis, and Russia's invasion of Ukraine, which threatens to disrupt agri-food supply chains [47–49]. Furthermore, in the case of agri-food products, the assessment included multiple layers, with food security being one of the most important [50,51]. Agriculture and the food industry are strategic economic sectors because they contribute to food security [29,52].

The study further explored the relationships between food insecurity and the aforementioned factors, including the RCA and LFI indices as explanatory variables with other macroeconomic variables. This can further expand the policy interventions toward enhancing food security as supported by international trade conditions for agricultural commodities. To achieve the objective, the yearly averages for the RCA and LFI for the selected agri-food commodities were computed for the analysis. In similar conditions, Sarris and Morrison [53] also utilized an averaged trade balance score across agricultural commodities in a market and trade policy analysis study to enhance African food security. Giuliani et al. [54] also adopted a similar computation technique in a global value chain analysis study using Latin American countries as case studies.

The aggregated Household Food Insecurity Access Scale (HFIAS) was adopted for the analysis in Namibia. The generic HFIAS was developed by Coates et al. [55], and a modified version of the HFIAS was also validated in Iran [56]. The tool is a nine-item scale with a reference period of the past four weeks for all included questions [57]. Using HFIAS, households were asked to respond to each experience as never, rarely, sometimes, or often, generating a total score from 0 to 27 [58]. A higher score indicates a higher level of household food insecurity. Equation (6) shows the standard procedure for calculating the HFIAS.

$$\text{HFIAS} = \sum_{i=1}^{n} X_i F_i \tag{6}$$

where "HFIAS" is the score, "$X_i$" is the food insecurity occurrence, and "$F_i$" is the frequency of the associated occurrence.

The construction of the HFIAS was based on an extensive review examining commonalities in the experience and expression of food insecurity (defined as lack of access) across cultures and locations in Namibia. This allowed the household-level index to be nationally aggregated and included in the model. Primary data in 2021 was used to generate and validate the HFIAS, while secondary data sources were used to estimate the HFIAS for

2000 to 2021. To achieve this, the weighted mean averages were computed from the widely reported food insecurity categories: food secure, mildly food insecure, moderately food insured, and severely food insecure. The median values for these categories were used and summed into the weighted meanwhile, using the total range (27) as the weighting factor (denominator), as shown in Equation (7).

$$
\begin{aligned}
\text{Aggregate Weighted HFIAS} &= \text{weighted HFIAS (food secure)} \\
&+ \text{weighted HFIAS (mildly food insecure)} \\
&+ \text{weighted HFIAS (moderately food insure)} \\
&+ \text{weighted HFIAS (severely food insecure)}
\end{aligned} \tag{7}
$$

This measure of food insecurity is also consistent with other food insecurity studies, such as Amare et al. [59] in Nigeria and Tabe-Ojong et al. [60] in Kenya, Namibia, and Tanzania.

The implications of food security of RCA and LFI were conducted using HFIAS as an outcome variable, as indicated in Figure 1 and Equations (8) and (9). It implies that RCA and LFI, macroeconomic stability, GDP per capita, land and labor productivity are explanatory variables.

$$
\text{HFIAS}_{it} = \alpha_0 + \alpha\,\text{RCA}_{it} + \alpha_1 \text{LAND}_{it} + \alpha_2\,\text{LABOUR}_{it} + \alpha_3\,\text{GDPpc}_{it} + \alpha_4\,\text{INF}_{it} + \varepsilon_i \tag{8}
$$

$$
\text{HFIAS}_{it} = \alpha_0 + \alpha\,\text{LFI}_{it} + \alpha_1 \text{LAND}_{it} + \alpha_2\,\text{LABOUR}_{it} + \alpha_3\,\text{GDPpc}_{it} + \alpha_4\,\text{INF}_{it} + \varepsilon_i \tag{9}
$$

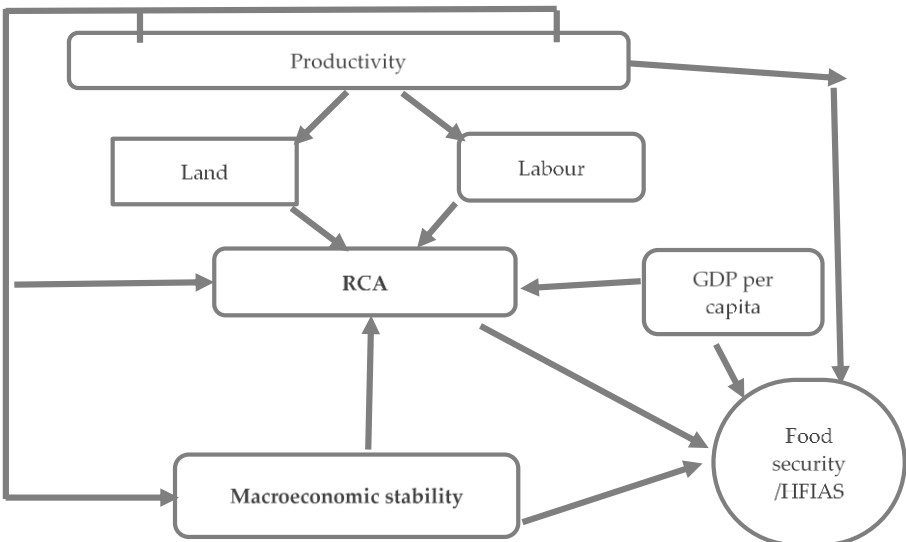

**Figure 1.** Conceptual framework of the determinants to the revealed comparative advantage (RCA).

In Figure 1, concepts are understood and developed based on the data patterns and not by collecting data to evaluate the hypothesis. Still, the study also emphasizes the research's validity through an empirical approximation. The model includes an RCA and LFI indicator as a dependent variable, while explanatory variables include land and labor productivity, GDP per capita, and macroeconomic stability. From the food security implication aspect, the model includes the HFIAS indicator as a dependent variable, while explanatory variables include RCA and LFI, land and labor productivity, GDP per capita, and macroeconomic stability.

This study contains several hypotheses based on the objective of the study, which is defined as follows:

**Hypothesis 1.** *Land productivity has a positive influence on Namibia's agri-food comparative advantages.*

**Hypothesis 2.** *Labor productivity has a positive influence on Namibia's agri-food comparative advantages.*

**Hypothesis 3.** *GDP per capita has a positive influence on Namibia's agri-food comparative advantages.*

**Hypothesis 4.** *Macroeconomic stability has a positive impact on Namibia's agri-food comparative advantages.*

**Hypothesis 5.** *RCA and LFI have a positive influence on Namibia's food security.*

**Hypothesis 6.** *Land productivity has a positive influence on Namibia's food security.*

**Hypothesis 7.** *Labor productivity has a positive influence on Namibia's food security.*

**Hypothesis 8.** *GDP per capita has a positive influence on Namibia's food security.*

**Hypothesis 9.** *Macroeconomic stability has a positive impact on Namibia's food security.*

*2.7. Determinants of RCA and LFI*

Relative comparative advantage (RCA) and Lafay Index (LFI) are fundamental principles in international trade and form the core foundation of the importance of free trade for agricultural commodities. Understanding the relationship between the RCA and LFI index and macroeconomic factors such as GDP and inflation can be helpful for policy design.

The impact of the determinants of RCA and LFI was conducted for 2000–2021. The outcome variable is RCA and LFI, while macroeconomic stability, GDP per capita, and land and labor productivity are independent variables. The theoretical framework of the model is depicted in Figure 1, and the model is empirically expressed in Equations (10) and (11):

$$RCA_{it} = \alpha_0 + \alpha_1 LAND_{it} + \alpha_2\ LABOUR_{it} + \alpha_3\ GDPpc_{it} + \alpha_4\ INF_{it} + \varepsilon_i \tag{10}$$

$$LFI_{it} = \alpha_0 + \alpha_1 LAND_{it} + \alpha_2\ LABOUR_{it} + \alpha_3\ GDPpc_{it} + \alpha_4\ INF_{it} + \varepsilon_i \tag{11}$$

**3. Results**

*3.1. Revealed Comparative Advantage (RCA) and Lafay Index (LFI) for Agri-Food Commodities*

The RCA for live animals (00), meat and meat preparations (01), fish (03), and raw hides (21) in Namibia revealed a comparative advantage during the study period from 2000 to 2021 (Tables A1 and A2 in Appendix A). This meant that Namibia was a net exporter of the above-mentioned agri-food commodities.

As Tables A1 and A2 indicate dairy (02), coffee (07), and cereals (04) except in 2002, miscellaneous (09), silk (26), and animal and vegetable oils (43) showed a comparative disadvantage during the study period of 2000–2021. However, vegetables and Fruit (05) and tobacco (12) showed mixed results. Vegetables and fruit (05) showed a comparative advantage from 2005–2010 and 2021, on the other hand, a comparative disadvantage for the rest of the study period of 2000–2004 and 2011–2020. Tobacco (12) showed a comparative advantage in three years 2000, 2017, and 2018, and a comparative disadvantage in the rest of the study period of 2001–2016 and 2019–2021. Bahta and Willems [10] pointed out that the RCA explains how a country fits into the global trade picture and that the RCA is based on relative export shares, which could be skewed due to trade and non-trade restrictions.

The LFI in Table A1 indicated fish (03) showed a significant comparative advantage. This meant that the higher the value of this index, the more specialized Namibia was in the fish trade. It was the fact that live animal (00) also showed a comparative advantage for 14 years (2001–2007, 2010–2011, 2015, 2014–2019) and showed a comparative disadvantage for the rest of the years (2000, 2008–2009, 2012–2014, 2016, 2020–2021).

*3.2. Export Diversification Index and Major Export Categories of Agri-Food Commodities*

The EDI values for all agri-food commodities (Tables A3–A5 in Appendix A) were close to zero, indicating a less concentrated trade structure and exhibited low concentrations throughout the study period, according to the index of trade concentration (HI). Lower

trade concentrations will lessen the impact of international trade risk due to the likelihood of price fluctuation in the international market.

From 2000 to 2021, the MEC for all agri-food commodities (Tables A3 and A4) was below 50%, indicating that Namibia was not considered overly reliant. According to Nin-Pratt et al., fish, grapes, beef and seafood, and fuelwood were Namibia's top agricultural export commodities [61].

### 3.3. Model Estimates and Economic Analysis

3.3.1. RCA and LFI as Outcome Variables

Multiple linear regression modeling was then used to estimate the relationships between the RCA, LFI, HFIAS, and selected independent variables of interest. The impact of macroeconomic stability, GDP per capita, and land and labor productivity on the RCA and LFI level of agri-food commodities during 2000–2021 was estimated. The economic analysis included descriptive statistics, diagnostics tests, and regression models.

The descriptive results are presented in Table A6 and indicate significant differences in the level of RCA for different agri-food commodities. The highest RCA indicator values were recorded for fish, with a 19.30 mean and a range of 13.20–31.65, followed by live animals, with an 18.38 mean and values ranging from 6.88 to 31.17. On the other hand, animal and vegetable oils were the lowest 0.06 mean and a range of 0.00–0.22. Regarding the explanatory variables, the highest value indicator was recorded for GDPpc with 4321.04 (Max) and the lowest with land productivity at 0.94 (Min).

Table A6 also indicated significant differences in the level of LFI for different agri-food commodities. The highest LFI indicator values were recorded for fish, with a 6.61 mean and a range of 4.73–11.88, followed by live animals, with a 1.06 mean and values ranging from 0.42 to 1.78. On the other hand, Cereals was the lowest −0.90 mean and a range of −1.13–(−0.68).

Table A6 shows the Skewness of RCA, LFI, and other variables. The RCA of Live animals, Meat and meat preparations, vegetables and fruit, coffee, tea, cocoa, spices, and manufacturers indicated that the data are fairly symmetrical. On the other hand, the LFI of Live animals, Meat and meat preparations, Dairy products and birds' eggs, Cereals and cereal preparations, Vegetables and fruit and Coffee, tea, cocoa, spices, and manufactures thereof indicated that the data are fairly symmetrical.

Kurtosis of the RCA of Live animals, Meat and meat preparations, Dairy products and birds' eggs, Fish, crustaceans, mollusks, and aquatic invertebrates and preparations thereof, Vegetables and fruit, Sugar, sugar preparations and honey, Coffee, tea, cocoa, spices, and manufactures thereof, Miscellaneous edible products and preparations indicated that the dataset has lighter tails than a normal distribution. On the other hand, Kurtosis of the LFI of Live animals, Meat and meat preparations, Dairy products and birds' eggs, Fish, crustaceans, mollusks, and aquatic invertebrates and preparations thereof, Cereals and cereal preparations, Vegetables and fruit, Coffee, tea, cocoa, spices, and manufactures thereof, Miscellaneous edible products and preparations, Tobacco, and raw hides and indicated that the dataset has lighter tails than a normal distribution.

The coefficient of variation varies from −68.87% for Vegetables and fruit to 125% for RCA hides and skins, indicating that for some commodities, the standard deviation is relatively less compared to the mean, and for other commodities standard deviation is somewhat more compared to the mean.

The Jarque–Bera test indicated that for commodity LFI Fish, RCA Cereals, RCA and LFI Sugar, RCA and LFI Miscellaneous edible products and preparations, RCA Tobacco, RCA, and LFI hide and skins, LFI Animal, vegetable fats and oils deviate from normality statistically significant at 1%. Other variables were significant at 5 and 10%, as indicated in Table A6.

More analysis of the multicollinearity test was conducted in aggregate regression analysis of RCA, LFI, and HFIAS as outcome variables, as explained in detail in Tables 1–4.

**Table 1.** Regression results for the RCA.

| Variable | Coefficient | t-Value | *p*-Value | VIF | 1/VIF |
|---|---|---|---|---|---|
| Labor | $-2.53 \times 10^{-5}$ ($6.77 \times 10^{-6}$) | −3.73 * | 0.002 | 1.07 | 0.93 |
| Land | −5.05 (1.560) | −3.24 * | 0.005 | 3.14 | 0.32 |
| GDPpc | $2.14 \times 10^{-4}$ ($1.687 \times 10^{-4}$) | 1.27 | 0.222 | 3.26 | 0.31 |
| Inflation | −0.086 (0.078) | −1.11 | 0.284 | 1.09 | 0.92 |
| Constant | 9.341 (1.591) | 5.87 * | 0.000 | Mean VIF: 2.14 | |
| Prob > F | 0.0001 * | | | | |
| R-squared | 0.5079 | | | | |
| F (4,17) | 12.62 | | | | |
| No. of obs. | 22 | | | | |

| White's test for Ho: homoscedasticity; against Ha: unrestricted heteroskedasticity |
|---|
| chi2(14) = 17.40 |
| Prob > chi2 = 0.2356 |

| Cameron and Trivedi's decomposition of IM-test | | | |
|---|---|---|---|
| Source | chi2 | df | p |
| Heteroskedasticity | 17.40 | 14 | 0.24 |
| Skewness | 7.40 | 4 | 0.12 |
| Kurtosis | 2.79 | 1 | 0.095 |
| Total | 27.59 | 19 | 0.092 |

Notes: * indicate *p*-values significant at 1% levels; t-values estimated on robust standard errors in parenthesis.

**Table 2.** Regression results for the LFI.

| Variable | Coefficient | t-Value | *p*-Value | VIF | 1/VIF |
|---|---|---|---|---|---|
| Labor | $-2.16 \times 10^{-7}$ ($1.52 \times 10^{-6}$) | −0.14 | 0.889 | 1.07 | 0.93 |
| Land | −0.6081 (0.253) | −2.40 ** | 0.028 | 3.14 | 0.32 |
| GDPpc | $-2.99 \times 10^{-5}$ ($4.06 \times 10^{-5}$) | −0.74 | 0.471 | 3.26 | 0.31 |
| Inflation | −0.011 (0.016) | −0.68 | 0.504 | 1.09 | 0.92 |
| Constant | 1.321 (0.249) | 5.30 * | 0.000 | Mean VIF: 2.14 | |
| Prob > F | 0.003 * | | | | |
| R-squared | 0.5229 | | | | |
| F (4,17) | 6.13 | | | | |
| No. of obs. | 22 | | | | |

| White's test for Ho: homoscedasticity; against Ha: unrestricted heteroskedasticity |
|---|
| chi2(14) = 18.33 |
| Prob > chi2 = 0.19 |

| Cameron & Trivedi's decomposition of IM-test | | | |
|---|---|---|---|
| Source | chi2 | df | p |
| Heteroskedasticity | 18.33 | 14 | 0.19 |
| Skewness | 5.25 | 4 | 0.262 |
| Kurtosis | 1.28 | 1 | 0.258 |
| Total | 24.86 | 19 | 0.17 |

Notes: * and ** indicate p-values significant at 1% and 5% levels, respectively; t-values estimated on robust standard errors in parenthesis.

Results in Table 1 show the relationship between the RCA and four independent variables: labor productivity, land productivity, GDP per capita, and inflation. Of these four variables, labor and land productivity significantly negatively influence the aggregated RCA for agri-food commodities. The Variance Inflation Factor (VIF) values are all less than 10, and there is strong evidence for not suspecting multicollinearity [62]. The mean VIF is 2.14. The White test for heteroscedasticity was also used in the study. From the results, the prob > chi2 = 0.24. The null hypothesis of constant variance cannot be rejected at a 5% significance level. The above finding implies no suspicion of heteroscedasticity in the residuals.

**Table 3.** Regression results for the HFIAS—RCA as explanatory Variable.

| Variable | Coefficient | t-Value | *p*-Value | VIF | 1/VIF |
|---|---|---|---|---|---|
| RCA | $-0.516$ (0.719) | $-0.72$ | 0.484 | 2.03 | 0.49 |
| Labor | $-6.37 \times 10^{-5}$ ($4.41 \times 10^{-5}$) | $-1.44$ | 0.168 | 1.11 | 0.90 |
| Land | $-18.429$ (8.682) | $-2.12$ ** | 0.050 | 4.88 | 0.20 |
| GDPpc | $2.632 \times 10^{-3}$ ($8.844 \times 10^{-3}$) | 2.98 * | 0.009 | 3.45 | 0.29 |
| Inflation | $-0.175$ (0.328) | $-0.53$ | 0.601 | 1.17 | 0.85 |
| Constant | 32.249 (8.868) | 3.64 * | 0.02 | Mean VIF: 2.53 | |
| Prob > F | 0.0071 * | | | | |
| R-squared | 0.3844 | | | | |
| F (5,16) | 4.81 | | | | |
| No. of obs. | 22 | | | | |

| White's test for Ho: homoscedasticity; against Ha: unrestricted heteroskedasticity |
|---|
| chi2(14) = 21.97 |
| Prob > chi2 = 0.34 |

| Cameron & Trivedi's decomposition of IM-test | | | |
|---|---|---|---|
| Source | chi2 | Df | p |
| Heteroskedasticity | 21.97 | 20 | 0.34 |
| Skewness | 9.22 | 5 | 0.10 |
| Kurtosis | 1.17 | 1 | 0.28 |
| Total | 32.36 | 26 | 0.18 |

Notes: * and ** indicate *p*-values significant at 1% and 5% levels, respectively; t-values estimated on robust standard errors in parenthesis.

**Table 4.** Regression results for the HFIAS-LFI as explanatory Variable.

| Variable | Coefficient | t-Value | *p*-Value | VIF | 1/VIF |
|---|---|---|---|---|---|
| LFI | 1.922 (2.415) | 0.80 | 0.438 | 2.10 | 0.49 |
| Labor | $-5.02 \times 10^{-5}$ ($3.97 \times 10^{-5}$) | $-1.27$ | 0.223 | 1.07 | 0.90 |
| Land | $-14.652$ (7.835) | $-1.87$ *** | 0.080 | 3.75 | 0.20 |
| GDPpc | $2.58 \times 10^{-5}$ ($9.132 \times 10^{-3}$) | 2.82 ** | 0.012 | 3.35 | 0.29 |
| Inflation | $-0.109$ (0.312) | $-0.35$ | 0.731 | 1.12 | 0.85 |
| Constant | 24.888 (6.346) | 3.92 * | 0.001 | Mean VIF: 2.28 | |
| Prob > F | 0.0001 * | | | | |
| R-squared | 0.3793 | | | | |
| F (5,16) | 10.48 | | | | |
| No. of obs. | 22 | | | | |

| White's test for Ho: homoscedasticity; against Ha: unrestricted heteroskedasticity |
|---|
| chi2(14) = 21.97 |
| Prob > chi2 = 0.36 |

| Cameron and Trivedi's decomposition of IM-test | | | |
|---|---|---|---|
| Source | chi2 | df | p |
| Heteroskedasticity | 21.97 | 20 | 0.36 |
| Skewness | 9.05 | 5 | 0.11 |
| Kurtosis | 0.73 | 1 | 0.39 |
| Total | 31.35 | 26 | 0.22 |

Notes: *; ** and *** indicate *p*-values significant at 1%, 5%, and 10% levels, respectively; t-values estimated on robust standard errors in parenthesis.

Table 2 shows the results for the relationship between the LFI and the four independent variables of interest. In this instance, the results imply a significant negative association between land productivity. Since the same variables were used as with the RCA model, the VIF conclusions also suggest no multicollinearity in the LFA model. From the results, the prob > chi2 = 0.19. The null hypothesis of constant variance cannot be rejected at

a 5% significance level. The above finding implies no suspicion of heteroscedasticity in the residuals.

### 3.3.2. HFIAS as Outcome Variables

The results in Tables 3 and 4 represent the relationship between the HFIAS and selected variables, including the RCA and LFI, respectively. Of the five variables, including RCA as an explanatory variable, only land productivity and GDP influence the extent of food insecurity in Namibia. The Variance Inflation Factor (VIF) values are all less than 10, and there is strong evidence for not suspecting multicollinearity [62]. The mean VIF is 2.53, and the prob > chi2 = 0.34. The null hypothesis of constant variance cannot be rejected at a 5% significance level. The above finding implies no suspicion of heteroscedasticity in the residuals.

Further, Table 4 indicates the relationship between the HFIAS and selected variables, including the LFI. Of the five variables, including LFI as an explanatory variable, evidence suggested that land productivity and the GDP per capita are also significant determinants of food security in Namibia. This implies that land productivity and GDP per capita influence the extent of food insecurity in Namibia. The Variance Inflation Factor (VIF) values are all less than 10, and there is strong evidence for not suspecting multicollinearity [62]. The mean VIF is 2.28, and the prob > chi2 = 0.36. The null hypothesis of constant variance cannot be rejected at a 5% significance level. The above finding implies no suspicion of heteroscedasticity in the residuals.

### 4. Discussion

The RCA and LFI of agri-food commodities during the study revealed mixed results. The RCA for live animals, meat and meat preparations, fish, and raw hides in Namibia showed a comparative advantage during the study period from 2000 to 2021. This implies that Namibia was a net exporter of the above-mentioned agri-food commodities. On the other hand, dairy, coffee, cereals, except in 2002, miscellaneous, silk, and animal and vegetable oils showed a comparative disadvantage during the study period of 2000–2021. However, vegetables and fruit, and tobacco showed mixed results. Vegetables and fruit showed a comparative advantage from 2005–2010 and 2021 and a comparative disadvantage the rest of the study period of 2000–2004 and 2011–2020. Tobacco showed a comparative advantage in three years, 2000, 2017, and 2018 and a comparative disadvantage in the rest of the study period of 2001–2016 and 2019–2021. This study concurred with Constantin et al. [63], which found that a competitiveness trade-off cereal chain from Ireland, the Netherlands, Belgium, and Portugal and the opposite was observed for Romania, Bulgaria, and Hungary.

Further, Taghouti et al. [64] found that the Tunisian export of fish is competitive. Regarding vegetables and cereals, Tunisian export is not competitive. However, their findings regarding live animals contradict this study's findings.

LFI indicated fish showed a significant comparative advantage. This implies that the higher the value of this index, the more specialized Namibia was in the fish trade. It was the fact that live animals also showed a comparative advantage for 14 years (2001–2007, 2010–2011, 2015, 2014–2019 and showed a comparative disadvantage for the rest of the years (2000, 2008–2009, 2012–2014, 2016, 2020–2021). The RCA and LFI of live animals and fish indicate that a comparative advantage implies a net export and greater specialization for these products. These findings concurred with Bahta and Willems [10], pointing out that the RCA explained how it fits into the global trade picture and that the RCA is based on relative export shares, which could be skewed due to trade non-trade restrictions.

The EDI and HI result shows that all agri-food commodities were close to zero, indicating a less concentrated trade structure. Further, the result attested that lower trade concentrations would lessen the impact of international trade risk due to the likelihood of price fluctuation in the global market. Furthermore, the MEC result confirms that all agri-food commodities were below 50%; this is interpreted as Namibia not being overly

reliant on the agri-food commodities mentioned above. These findings agreed with Nin-Pratt et al. [61], who discovered that Namibia's top agricultural export goods were fish, grapes, beef, and non-alcoholic beverage.

The regression result indicated that labor and land productivity significantly negatively influence the aggregated RCA, and land productivity has a significant negative relationship with the aggregated LFI for agri-food commodities. When looking at the implication of RCA and LFI on food security, of the five variables, only land productivity and GDP per capita influence the extent of food insecurity in Namibia. This implies that Namibia's agri-food industry had a less concentrated trade structure and did not depend on international trade from the agri-food industry and does not have much implication for food security. These evaluations provide policymakers with information on agri-food competitiveness, factors that influence the industry's competitiveness, and implications for food security. These findings aligned with the results of Hamulczuk and Pawlak [65]; they proved that increasing trade openness and relative demand have a positive impact on the international competitiveness of the food industry and food security worldwide. Besides, European Commission [66] highlighted that, in the agri-food sector, productivity, including land productivity is the most reliable indicator of competitiveness in the long term to enhance food security. Furthermore, Patel-Campillo [67] indicated that agri-export specialization (competitiveness) increases food security by fostering economic growth through foreign exchange earnings and increased employment.

The findings revealed that agricultural productivity was critical in defining Namibia's agri-food sector and competitiveness. Increased production and productivity are crucial to meet the ever-increasing demand for food and agri-food products while enhancing income and competitiveness. Increased competitiveness in countries such as Namibia is critical because many people rely on agriculture, and smallholder farming is sustained; as a result, poverty and hunger are reduced, thereby achieving the SDGs 2030 agenda for ending hunger and poverty [35,68]. The study findings contradict those of Matkovski et al. [35] and Jambor and Babu [69], who discovered a negative relationship between GDP per capita and agri-food competitiveness. Further, the findings against the study by Bahta [21], who found that GDP per capita significantly positively influences comparative advantage besides productivity.

## 5. Conclusions

The SDGs, adopted by the member States of the United Nations in September 2015, show that agri-food trade is essential to achieve sustainable food and nutrition security. Different drivers (at global, domestic, and local levels) have changed the structure and functioning of the agri-food trade. Agri-food trade means both challenges and opportunities for the actors of the agri-food value chains. It also has implications in terms of food security. Indeed, the factor determining the competitiveness of agri-food affects food security. Therefore, attaining long-term food security means understanding the dynamics of global agri-food trade as well as the governance and functioning of domestic agri-food markets is crucial. In fact, agri-food markets ought to be managed to increase the advantages of widened access to markets while mitigating the risks related to higher exposure to international competition and market volatility. As highlighted by Constantin et al. [63], to achieve SDGs divergent action is required such as strategies designed to capitalize on sustainable economic competitiveness. Agricultural sustainable economic competitiveness can be achieved through a mix of strategic actions of decision-makers, which needs to be grounded in a solid factor endowment foundation and efficient trade specialization policy.

This research aims to determine the competitiveness of Namibian agri-food products, the factors that influence them, and the implication for food security. The study revealed: (i) a mixed result of RCA and LFI of agri-food commodities, (ii) a less concentrated trading structure, (iii) Namibia was not reliant on overseas commerce from the agri-food sector, (iv) labor and land productivity has a significant negative influence on the aggregated RCA and LFI, and (v) land productivity and GDP per capita influence the extent of food inse-

curity in Namibia. This implies the importance of agricultural productivity to Namibia's agri-food competitiveness. As a result, the Namibia government, with stakeholders in the industry, should strengthen the agri-food sector's competitiveness to endure global market pressure. Instead of exporting value-added products, Namibia could reduce its reliance on imported raw materials by developing a competitive food industry, enhancing productivity, and enhancing the performance of supporting industries and logistics infrastructure. Productivity, including labor and land in the agri-food sector, is a vector of competitiveness that should be empowered starting from primary agriculture and should be harnessed. Raw agricultural materials can provide strategic competitiveness if they are used in processing activities and then generate high levels of comparative advantage as a result of beneficial trade flows. This means that Namibia's future agriculture and agri-food policies should prioritize the development of a modern agribusiness industry. Trade-based measures of competitiveness provide a realistic indicator of underlying competitiveness only in the absence of significant barriers to trade, the impact of domestic market influence was a limitation. However, the study believes the results are reliable enough to draw agri-food policy recommendations. The study recommends that future research should incorporate trade and non-trade barriers and domestic market influence, which was beyond the scope of this study.

**Author Contributions:** All authors significantly contributed to the present manuscript preparation. S.M. was involved in data collection, analysis, and writing the first draft. Y.T.B. aided in the study design and conceptualization, review, and writing the final draft. All authors have read and agreed to the published version of the manuscript.

**Funding:** This research received no external funding.

**Data Availability Statement:** Data will be available on request from the corresponding author (Y.T.B.).

**Conflicts of Interest:** The authors declare no conflict of interest.

## Appendix A

**Table A1.** Revealed Comparative Advantage (RCA) and Lafay Index (LFI) for food and live animals during 2000–2021.

| Year | Live Animals (00) | | Meat & M. Preparations (01) | | Fish (03) | | Dairy (02) | | Vegetables & Fruit (05) | | Sugar & Honey (06) | | Coffee (07) | |
|---|---|---|---|---|---|---|---|---|---|---|---|---|---|---|
| | RCA | LFI | RCA | LFI | RCA | LFI | RCA | LFI | RCA | LFI | RCA | LFI | RCA | LFI |
| 2000 | 6.88 | 0.42 | 2.63 | −0.02 | 23.31 | 8.50 | 0.14 | −0.40 | 0.44 | −0.91 | 1.64 | −0.81 | 0.17 | −0.34 |
| 2001 | 22.91 | 1.52 | 3.42 | 0.54 | 27.05 | 10.83 | 0.18 | −0.28 | 0.41 | −0.63 | 1.57 | −0.59 | 0.30 | −0.25 |
| 2002 | 15.59 | 1.21 | 2.80 | 0.45 | 26.01 | 10.26 | 0.36 | −0.19 | 0.60 | −0.39 | 3.34 | −0.45 | 0.18 | −0.21 |
| 2003 | 24.19 | 1.61 | 5.10 | 1.24 | 31.65 | 11.88 | 0.44 | −0.25 | 0.98 | −0.41 | 3.71 | −0.39 | 0.19 | −0.32 |
| 2004 | 22.89 | 1.40 | 5.77 | 1.30 | 22.13 | 7.54 | 0.26 | −0.39 | 0.95 | −0.75 | 1.72 | −0.74 | 0.30 | −0.46 |
| 2005 | 29.40 | 1.72 | 7.45 | 1.89 | 21.64 | 7.11 | 0.16 | −0.34 | 1.08 | −0.63 | 1.31 | −0.83 | 0.37 | −0.35 |
| 2006 | 24.09 | 1.33 | 5.89 | 1.21 | 19.58 | 6.02 | 0.14 | −0.37 | 1.11 | −0.44 | 0.65 | −0.87 | 0.30 | −0.37 |
| 2007 | 21.91 | 1.15 | 5.64 | 1.31 | 18.32 | 5.26 | 0.15 | −0.31 | 1.17 | −0.35 | 0.65 | −1.59 | 0.29 | −0.24 |
| 2008 | 10.83 | 0.54 | 4.90 | 0.97 | 19.08 | 5.18 | 0.13 | −0.31 | 1.00 | −0.39 | 0.90 | −0.60 | 0.25 | −0.22 |
| 2009 | 10.48 | 0.63 | 4.45 | 1.06 | 16.67 | 5.45 | 0.17 | −0.30 | 1.06 | −0.32 | 0.57 | −0.78 | 0.22 | −0.23 |
| 2010 | 18.71 | 1.02 | 4.65 | 0.96 | 19.32 | 6.10 | 0.27 | −0.30 | 1.05 | −0.41 | 0.63 | −0.85 | 0.20 | −0.28 |
| 2011 | 22.52 | 1.20 | 4.50 | 0.96 | 19.00 | 5.99 | 0.17 | −0.31 | 0.72 | −0.57 | 0.75 | −0.72 | 0.18 | −0.25 |
| 2012 | 13.32 | 0.72 | 4.76 | 1.31 | 20.41 | 6.43 | 0.13 | −0.30 | 0.83 | −0.42 | 0.56 | −0.63 | 0.28 | −0.17 |
| 2013 | 16.01 | 0.86 | 3.20 | 0.93 | 17.53 | 8.37 | 0.08 | −0.29 | 0.66 | −0.42 | 0.32 | −0.60 | 0.16 | −0.18 |
| 2014 | 9.51 | 0.51 | 2.52 | 0.70 | 15.91 | 5.44 | 0.09 | −0.24 | 0.63 | −0.30 | 0.21 | −0.44 | 0.01 | −0.19 |
| 2015 | 22.33 | 1.20 | 2.86 | 0.72 | 17.33 | 5.91 | 0.04 | −0.25 | 0.66 | −0.28 | 0.19 | −0.49 | 0.02 | −0.21 |
| 2016 | 11.75 | 0.66 | 1.94 | 0.44 | 15.7 | 5.95 | 0.01 | −0.28 | 0.55 | −0.35 | 0.11 | −0.59 | 0.01 | −0.24 |
| 2017 | 31.17 | 1.78 | 1.94 | 0.36 | 16.67 | 6.22 | 0.08 | −0.30 | 0.77 | −0.19 | 0.05 | −0.63 | 0.09 | −0.44 |
| 2018 | 24.56 | 1.37 | 1.39 | 0.19 | 13.20 | 4.73 | 0.09 | −0.24 | 0.78 | −0.12 | 0.07 | −0.47 | 0.09 | −0.37 |
| 2019 | 19.04 | 1.08 | 1.85 | 0.41 | 13.85 | 4.96 | 0.06 | −0.23 | 0.93 | −0.02 | 0.09 | −0.49 | 0.09 | −0.34 |
| 2020 | 11.97 | 0.70 | 1.11 | 0.01 | 13.96 | 4.93 | 0.05 | −0.25 | 0.89 | 0.07 | 0.04 | −0.60 | 0.03 | −0.37 |
| 2021 | 14.39 | 0.70 | 1.18 | 0.01 | 15.53 | 4.93 | 0.05 | −0.25 | 1.05 | 0.07 | 0.04 | −0.60 | 0.03 | −0.37 |

**Table A2.** Revealed Comparative Advantage (RCA) and Lafay Index (LFI) for food, Tobacco, crude materials, and animal vegetables during 2000–2021.

| | Food | | | | Tobacco | | Crude Materials | | | | Animal & Vegetable Oils | |
|---|---|---|---|---|---|---|---|---|---|---|---|---|
| | Cereals (04) | | Miscellaneous (09) | | Tobacco (12) | | Raw Hides (21) | | Silk (26) | | An. & Veg. Oils (43) | |
| Year | RCA | LFI | RCA | LFI | RCA | LFI | RCA | LFI | RCA | LFI | RCA | LFI |
| 2000 | 0.17 | −1.01 | 0.30 | −0.92 | 1.38 | −0.01 | 7.57 | 0.35 | 0.04 | −0.02 | 0.00 | −0.03 |
| 2001 | 0.26 | −0.94 | 0.20 | −0.56 | 0.35 | −0.05 | 4.65 | 0.25 | 0.28 | 0.03 | 0.07 | −0.02 |
| 2002 | 1.01 | −0.92 | 0.41 | −0.31 | 0.82 | 0.01 | 5.57 | 0.29 | 0.15 | −0.43 | 0.14 | −0.04 |
| 2003 | 0.69 | −0.94 | 0.81 | −0.28 | 0.75 | −0.1 | 5.28 | 0.24 | 0.37 | −0.02 | 0.07 | −0.05 |
| 2004 | 0.33 | −1.13 | 0.22 | −0.61 | 0.43 | −0.43 | 2.43 | 0.09 | 0.21 | 0.01 | 0.02 | −0.09 |
| 2005 | 0.34 | −0.87 | 0.22 | −0.72 | 0.55 | −0.42 | 3.25 | 0.09 | 0.22 | 0.01 | 0.02 | −0.04 |
| 2006 | 0.23 | −0.94 | 0.16 | −0.64 | 0.64 | −0.40 | 3.73 | 0.12 | 0.07 | −0.01 | 0.02 | −0.01 |
| 2007 | 0.15 | −0.94 | 0.18 | −0.48 | 0.24 | −0.35 | 4.86 | 0.12 | 0.06 | 0.00 | 0.01 | 0.00 |
| 2008 | 0.18 | −0.94 | 0.14 | −0.45 | 4.53 | 0.07 | 3.61 | 0.07 | 0.06 | −0.01 | 0.03 | −0.01 |
| 2009 | 0.25 | −0.94 | 0.16 | −0.52 | 1.07 | −0.35 | 2.93 | 0.07 | 0.03 | −0.01 | 0.02 | 0.00 |
| 2010 | 0.16 | −0.87 | 0.13 | −0.53 | 1.19 | −0.35 | 2.51 | 0.07 | 0.02 | −0.01 | 0.01 | 0.00 |
| 2011 | 0.14 | −0.90 | 0.15 | −0.5 | 0.28 | −0.40 | 2.89 | 0.09 | 0.02 | −0.01 | 0.01 | 0.00 |
| 2012 | 0.17 | −0.79 | 0.20 | −0.43 | 0.05 | −0.37 | 3.36 | 0.11 | 0.02 | −0.01 | 0.02 | 0.00 |
| 2013 | 0.22 | −0.93 | 0.22 | −0.42 | 0.21 | −0.33 | 2.79 | 0.11 | 0.01 | −0.01 | 0.00 | 0.00 |
| 2014 | 0.19 | −0.70 | 0.28 | −0.32 | 0.34 | −0.25 | 2.04 | 0.06 | 0.01 | −0.01 | 0.01 | 0.00 |
| 2015 | 0.42 | −0.68 | 0.38 | −0.32 | 0.34 | −0.26 | 1.20 | 0.03 | 0.01 | −0.01 | 0.00 | 0.00 |
| 2016 | 0.21 | −0.96 | 0.47 | −0.36 | 0.38 | −0.24 | 1.66 | 0.03 | 0.01 | −0.01 | 0.00 | 0.00 |
| 2017 | 0.16 | −0.84 | 0.39 | −0.5 | 1.29 | −0.27 | 3.04 | 0.02 | 0.03 | −0.01 | 0.14 | −0.02 |
| 2018 | 0.14 | −0.77 | 0.25 | −0.41 | 1.26 | −0.22 | 4.39 | 0.04 | 0.01 | 0.00 | 0.13 | −0.01 |
| 2019 | 0.17 | −0.93 | 0.29 | −0.40 | 0.86 | −0.21 | 4.20 | 0.03 | 0.03 | −0.01 | 0.22 | 0.00 |
| 2020 | 0.12 | −0.91 | 0.30 | −0.43 | 0.34 | −0.28 | 1.93 | 0.01 | 0.02 | −0.01 | 0.16 | −0.01 |
| 2021 | 0.12 | −0.91 | 0.33 | −0.43 | 0.40 | −0.28 | 1.81 | 0.01 | 0.02 | −0.01 | 0.14 | −0.01 |

**Table A3.** Export diversification and major export categories for food and live animals.

| | Live Animals (00) | | | Meat and Meat Preparations (01) | | | Fish (03) | | | Dairy (02) | | | Edible Vegetables & Fruit (05) | | |
|---|---|---|---|---|---|---|---|---|---|---|---|---|---|---|---|
| Year | MEC (%) | EDI | HI | MEC (%) | EDI | HI | MEC (%) | EDI | HI | MEC (%) | EDI | HI | MEC (%) | EDI | HI |
| 2000 | 0.0096 | 0.0048 | 0.0001 | 0.0183 | 0.0092 | 0.0003 | 0.1841 | 0.0920 | 0.0339 | 0.0006 | 0.0003 | 0.0000 | 0.0047 | 0.0024 | 0.0000 |
| 2001 | 0.0318 | 0.0159 | 0.0010 | 0.0262 | 0.0131 | 0.0007 | 0.2249 | 0.1124 | 0.0506 | 0.0008 | 0.0004 | 0.0000 | 0.0048 | 0.0024 | 0.0000 |
| 2002 | 0.0251 | 0.0125 | 0.0006 | 0.0208 | 0.0104 | 0.0004 | 0.2111 | 0.1055 | 0.0445 | 0.0016 | 0.0008 | 0.0000 | 0.0072 | 0.0036 | 0.0001 |
| 2003 | 0.0332 | 0.0166 | 0.0011 | 0.0376 | 0.0188 | 0.0014 | 0.2446 | 0.1223 | 0.0598 | 0.0020 | 0.0010 | 0.0000 | 0.0120 | 0.006 | 0.0001 |
| 2004 | 0.0287 | 0.0144 | 0.0008 | 0.0410 | 0.0205 | 0.0017 | 0.1581 | 0.0790 | 0.0250 | 0.0012 | 0.0006 | 0.0000 | 0.0106 | 0.0053 | 0.0001 |
| 2005 | 0.0354 | 0.0177 | 0.0013 | 0.0541 | 0.0271 | 0.0029 | 0.1505 | 0.0752 | 0.0227 | 0.0007 | 0.0003 | 0.0000 | 0.0120 | 0.006 | 0.0001 |
| 2006 | 0.0277 | 0.0139 | 0.0008 | 0.0390 | 0.0195 | 0.0015 | 0.1288 | 0.0644 | 0.0166 | 0.0006 | 0.0003 | 0.0000 | 0.0117 | 0.0058 | 0.0001 |
| 2007 | 0.0239 | 0.012 | 0.0006 | 0.0380 | 0.019 | 0.0014 | 0.1136 | 0.0568 | 0.0129 | 0.0007 | 0.0003 | 0.0000 | 0.0127 | 0.0063 | 0.0002 |
| 2008 | 0.0116 | 0.0058 | 0.0001 | 0.0356 | 0.0178 | 0.0013 | 0.1116 | 0.0558 | 0.0125 | 0.0006 | 0.0003 | 0.0000 | 0.0107 | 0.0053 | 0.0001 |
| 2009 | 0.0138 | 0.0069 | 0.0002 | 0.0372 | 0.0186 | 0.0014 | 0.1179 | 0.0589 | 0.0139 | 0.0008 | 0.0004 | 0.0000 | 0.0137 | 0.0068 | 0.0002 |
| 2010 | 0.0213 | 0.0106 | 0.0005 | 0.0350 | 0.0175 | 0.0012 | 0.1299 | 0.0649 | 0.0169 | 0.0012 | 0.0006 | 0.0000 | 0.0125 | 0.0062 | 0.0002 |
| 2011 | 0.0243 | 0.0121 | 0.0006 | 0.0341 | 0.017 | 0.0012 | 0.1267 | 0.0634 | 0.0161 | 0.0008 | 0.0004 | 0.0000 | 0.0082 | 0.0041 | 0.0001 |
| 2012 | 0.0149 | 0.0075 | 0.0002 | 0.0361 | 0.018 | 0.0013 | 0.1364 | 0.0682 | 0.0186 | 0.0006 | 0.0003 | 0.0000 | 0.0092 | 0.0046 | 0.0001 |
| 2013 | 0.0176 | 0.0088 | 0.0003 | 0.0247 | 0.0124 | 0.0006 | 0.1211 | 0.0605 | 0.0147 | 0.0004 | 0.0002 | 0.0000 | 0.0079 | 0.004 | 0.0001 |
| 2014 | 0.0113 | 0.0056 | 0.0001 | 0.0207 | 0.0104 | 0.0004 | 0.1176 | 0.0588 | 0.0138 | 0.0004 | 0.0002 | 0.0000 | 0.0079 | 0.0039 | 0.0001 |
| 2015 | 0.0262 | 0.0131 | 0.0007 | 0.0234 | 0.0117 | 0.0005 | 0.1312 | 0.0656 | 0.0172 | 0.0002 | 0.0001 | 0.0000 | 0.0092 | 0.0046 | 0.0001 |
| 2016 | 0.0142 | 0.0071 | 0.0002 | 0.0163 | 0.0081 | 0.0003 | 0.1306 | 0.0653 | 0.0171 | 0.0000 | 0.0000 | 0.0000 | 0.0083 | 0.0042 | 0.0001 |
| 2017 | 0.0366 | 0.0183 | 0.0013 | 0.0165 | 0.0082 | 0.0003 | 0.1335 | 0.0668 | 0.0178 | 0.0004 | 0.0002 | 0.0000 | 0.0095 | 0.0047 | 0.0001 |
| 2018 | 0.0278 | 0.0139 | 0.0008 | 0.0112 | 0.0056 | 0.0001 | 0.1020 | 0.0510 | 0.0104 | 0.0004 | 0.0002 | 0.0000 | 0.0089 | 0.0044 | 0.0001 |
| 2019 | 0.0233 | 0.0111 | 0.0005 | 0.0162 | 0.0081 | 0.0003 | 0.1076 | 0.0538 | 0.0116 | 0.0003 | 0.0001 | 0.0000 | 0.0110 | 0.0055 | 0.0001 |
| 2020 | 0.0145 | 0.0072 | 0.0002 | 0.0103 | 0.0052 | 0.0001 | 0.1079 | 0.0539 | 0.0116 | 0.0002 | 0.0001 | 0.0000 | 0.0116 | 0.0058 | 0.0001 |
| 2021 | 0.0145 | 0.0072 | 0.0002 | 0.0103 | 0.0052 | 0.0001 | 0.1079 | 0.0539 | 0.0116 | 0.0002 | 0.0001 | 0.0000 | 0.0116 | 0.0058 | 0.0001 |

Note: MEC: major export category; EDI: export diversification index; HI: Hirschman index.

**Table A4.** Export diversification and major export categories for food and live animals.

| Year | Coffee (07) | | | Cereals (04) | | | Sugar & Honey (06) | | | Miscellaneous (09) | | |
|---|---|---|---|---|---|---|---|---|---|---|---|---|
| | MEC (%) | EDI | HI | MEC (%) | EDI | HI | MEC (%) | EDI | HI | MEC (%) | EDI | HI |
| 2000 | 0.0008 | 0.0004 | 0.0000 | 0.0014 | 0.0007 | 0.0000 | 0.0035 | 0.0018 | 0.0000 | 0.0009 | 0.0004 | 0.0000 |
| 2001 | 0.0013 | 0.0006 | 0.0000 | 0.0023 | 0.0012 | 0.0000 | 0.0041 | 0.0021 | 0.0000 | 0.0007 | 0.0003 | 0.0000 |
| 2002 | 0.0009 | 0.0004 | 0.0000 | 0.0092 | 0.0046 | 0.0001 | 0.0085 | 0.0042 | 0.0001 | 0.0014 | 0.0007 | 0.0000 |
| 2003 | 0.0009 | 0.0004 | 0.0000 | 0.0060 | 0.003 | 0.0000 | 0.0091 | 0.0045 | 0.0001 | 0.0028 | 0.0014 | 0.0000 |
| 2004 | 0.0013 | 0.0007 | 0.0000 | 0.0027 | 0.0014 | 0.0000 | 0.0038 | 0.0019 | 0.0000 | 0.0007 | 0.0004 | 0.0000 |
| 2005 | 0.0015 | 0.0008 | 0.0000 | 0.0026 | 0.0013 | 0.0000 | 0.0030 | 0.0015 | 0.0000 | 0.0007 | 0.0004 | 0.0000 |
| 2006 | 0.0012 | 0.0006 | 0.0000 | 0.0017 | 0.0008 | 0.0000 | 0.0016 | 0.0008 | 0.0000 | 0.0005 | 0.0003 | 0.0000 |
| 2007 | 0.0012 | 0.0006 | 0.0000 | 0.0013 | 0.0007 | 0.0000 | 0.0014 | 0.0007 | 0.0000 | 0.0006 | 0.0003 | 0.0000 |
| 2008 | 0.0011 | 0.0006 | 0.0000 | 0.0018 | 0.0009 | 0.0000 | 0.0019 | 0.0009 | 0.0000 | 0.0005 | 0.0002 | 0.0000 |
| 2009 | 0.0012 | 0.0006 | 0.0000 | 0.0027 | 0.0013 | 0.0000 | 0.0016 | 0.0008 | 0.0000 | 0.0006 | 0.0003 | 0.0000 |
| 2010 | 0.0011 | 0.0005 | 0.0000 | 0.0015 | 0.0007 | 0.0000 | 0.0019 | 0.0009 | 0.0000 | 0.0004 | 0.0002 | 0.0000 |
| 2011 | 0.0010 | 0.0005 | 0.0000 | 0.0014 | 0.0007 | 0.0000 | 0.0022 | 0.0011 | 0.0000 | 0.0005 | 0.0003 | 0.0000 |
| 2012 | 0.0015 | 0.0007 | 0.0000 | 0.0018 | 0.0009 | 0.0000 | 0.0016 | 0.0008 | 0.0000 | 0.0007 | 0.0004 | 0.0000 |
| 2013 | 0.0008 | 0.0004 | 0.0000 | 0.0023 | 0.0011 | 0.0000 | 0.0009 | 0.0004 | 0.0000 | 0.0008 | 0.0004 | 0.0000 |
| 2014 | 0.0005 | 0.0003 | 0.0000 | 0.0020 | 0.001 | 0.0000 | 0.0005 | 0.0003 | 0.0000 | 0.0011 | 0.0006 | 0.0000 |
| 2015 | 0.0001 | 0.0001 | 0.0000 | 0.0045 | 0.0022 | 0.0000 | 0.0005 | 0.0002 | 0.0000 | 0.0016 | 0.0008 | 0.0000 |
| 2016 | 0.0001 | 0.0000 | 0.0000 | 0.0023 | 0.0011 | 0.0000 | 0.0003 | 0.0002 | 0.0000 | 0.0021 | 0.0011 | 0.0000 |
| 2017 | 0.0006 | 0.0003 | 0.0000 | 0.0015 | 0.0008 | 0.0000 | 0.0001 | 0.0001 | 0.0000 | 0.0029 | 0.0014 | 0.0000 |
| 2018 | 0.0005 | 0.0003 | 0.0000 | 0.0013 | 0.0006 | 0.0000 | 0.0002 | 0.0001 | 0.0000 | 0.0019 | 0.0009 | 0.0000 |
| 2019 | 0.0005 | 0.0003 | 0.0000 | 0.0017 | 0.0008 | 0.0000 | 0.0002 | 0.0001 | 0.0000 | 0.0023 | 0.0011 | 0.0000 |
| 2020 | 0.0002 | 0.0001 | 0.0000 | 0.0013 | 0.0006 | 0.0000 | 0.0001 | 0.0000 | 0.0000 | 0.0026 | 0.0013 | 0.0000 |
| 2021 | 0.0002 | 0.0001 | 0.0000 | 0.0013 | 0.0006 | 0.0000 | 0.0001 | 0.0000 | 0.0000 | 0.0026 | 0.0013 | 0.0000 |

**Table A5.** Export diversification and major export categories tobacco, crude material, and vegetables.

| Year | Tobacco | | | Crude Materials | | | | | | Animal and Vegetable Oils | | |
|---|---|---|---|---|---|---|---|---|---|---|---|---|
| | Tobacco (12) | | | Raw Hides (21) | | | Silk (26) | | | Animal, Vegetable Fats (43) | | |
| | MEC (%) | EDI | HI | MEC (%) | EDI | HI | MEC (%) | EDI | HI | MEC (%) | EDI | HI |
| 2000 | 0.0046 | 0.0023 | 0.0000 | 0.0076 | 0.0038 | 0.0001 | 0.0001 | 0.0001 | 0.0000 | 0.0000 | 0.0000 | 0.0000 |
| 2001 | 0.0012 | 0.0006 | 0.0000 | 0.0054 | 0.0027 | 0.0000 | 0.0009 | 0.0004 | 0.0000 | 0.0000 | 0.0000 | 0.0000 |
| 2002 | 0.0027 | 0.0014 | 0.0000 | 0.0060 | 0.003 | 0.0000 | 0.0004 | 0.0002 | 0.0000 | 0.0001 | 0.0000 | 0.0000 |
| 2003 | 0.0022 | 0.0011 | 0.0000 | 0.0052 | 0.0026 | 0.0000 | 0.0011 | 0.0005 | 0.0000 | 0.0000 | 0.0000 | 0.0000 |
| 2004 | 0.0011 | 0.0006 | 0.0000 | 0.0021 | 0.0011 | 0.0000 | 0.0006 | 0.0003 | 0.0000 | 0.0000 | 0.0000 | 0.0000 |
| 2005 | 0.0014 | 0.0007 | 0.0000 | 0.0025 | 0.0013 | 0.0000 | 0.0005 | 0.0003 | 0.0000 | 0.0000 | 0.0000 | 0.0000 |
| 2006 | 0.0014 | 0.0007 | 0.0000 | 0.0028 | 0.0014 | 0.0000 | 0.0002 | 0.0001 | 0.0000 | 0.0000 | 0.0000 | 0.0000 |
| 2007 | 0.0005 | 0.0003 | 0.0000 | 0.0031 | 0.0016 | 0.0000 | 0.0001 | 0.0001 | 0.0000 | 0.0000 | 0.0000 | 0.0000 |
| 2008 | 0.0098 | 0.0049 | 0.0001 | 0.0020 | 0.001 | 0.0000 | 0.0001 | 0.0001 | 0.0000 | 0.0000 | 0.0000 | 0.0000 |
| 2009 | 0.0030 | 0.0015 | 0.0000 | 0.0016 | 0.0008 | 0.0000 | 0.0001 | 0.0001 | 0.0000 | 0.0000 | 0.0000 | 0.0000 |
| 2010 | 0.0028 | 0.0014 | 0.0000 | 0.0017 | 0.0009 | 0.0000 | 0.0001 | 0.0000 | 0.0000 | 0.0000 | 0.0000 | 0.0000 |
| 2011 | 0.0006 | 0.0003 | 0.0000 | 0.0021 | 0.001 | 0.0000 | 0.0001 | 0.0000 | 0.0000 | 0.0000 | 0.0000 | 0.0000 |
| 2012 | 0.0001 | 0.0001 | 0.0000 | 0.0025 | 0.0013 | 0.0000 | 0.0000 | 0.0000 | 0.0000 | 0.0000 | 0.0000 | 0.0000 |
| 2013 | 0.0005 | 0.0002 | 0.0000 | 0.0025 | 0.0012 | 0.0000 | 0.0000 | 0.0000 | 0.0000 | 0.0000 | 0.0000 | 0.0000 |
| 2014 | 0.0008 | 0.0004 | 0.0000 | 0.0015 | 0.0008 | 0.0000 | 0.0000 | 0.0000 | 0.0000 | 0.0000 | 0.0000 | 0.0000 |
| 2015 | 0.0008 | 0.0004 | 0.0000 | 0.0009 | 0.0004 | 0.0000 | 0.0000 | 0.0000 | 0.0000 | 0.0000 | 0.0000 | 0.0000 |
| 2016 | 0.0009 | 0.0005 | 0.0000 | 0.0009 | 0.0004 | 0.0000 | 0.0000 | 0.0000 | 0.0000 | 0.0000 | 0.0000 | 0.0000 |
| 2017 | 0.0030 | 0.0015 | 0.0000 | 0.0007 | 0.0004 | 0.0000 | 0.0001 | 0.0000 | 0.0000 | 0.0001 | 0.0001 | 0.0000 |
| 2018 | 0.0029 | 0.0015 | 0.0000 | 0.0008 | 0.0004 | 0.0000 | 0.0000 | 0.0000 | 0.0000 | 0.0001 | 0.0001 | 0.0000 |
| 2019 | 0.0021 | 0.001 | 0.0000 | 0.0007 | 0.0003 | 0.0000 | 0.0001 | 0.0000 | 0.0000 | 0.0002 | 0.0001 | 0.0000 |
| 2020 | 0.0008 | 0.0004 | 0.0000 | 0.0002 | 0.0001 | 0.0000 | 0.0000 | 0.0000 | 0.0000 | 0.0002 | 0.0001 | 0.0000 |
| 2021 | 0.0008 | 0.0004 | 0.0000 | 0.0002 | 0.0001 | 0.0000 | 0.0000 | 0.0000 | 0.0000 | 0.0002 | 0.0001 | 0.0000 |

**Table A6.** Descriptive statistics of the RCA and LFI.

| | Outcome Variable—RCA and LFI | | | | | | | | | | | | | | | | | | | | | | | | | |
| | Food and Live Animals | | | | | | | | | | | | | | | | | | Tobacco | | Crude Materials | | | | An. & Veg. Oils | |
| | 00 | | 01 | | 02 | | 03 | | 04 | | 05 | | 06 | | 07 | | 09 | | 12 | | 21 | | 26 | | 43 | |
| | RCA | LFI | RCA | LFI | RCA | LFI | RCA | LFI | RCA | LFI | RCA | LFI | RCA | LFI | RCA | LFI | RCA | LFI | RCA | LFI | RCA | LFI | RCA | LFI | RCA | LFI |
|---|---|---|---|---|---|---|---|---|---|---|---|---|---|---|---|---|---|---|---|---|---|---|---|---|---|---|
| Mean | 18.38 | 1.06 | 3.63 | 0.77 | 0.15 | −0.29 | 19.30 | 6.61 | 0.27 | −0.90 | 0.83 | −0.37 | 0.87 | −0.67 | 0.18 | −0.29 | 0.28 | −0.48 | 0.80 | −0.25 | 3.44 | 0.10 | 0.08 | −0.02 | 0.06 | −0.02 |
| Stan. Error | 1.43 | 0.09 | 0.38 | 0.11 | 0.02 | 0.01 | 0.98 | 0.43 | 0.04 | 0.02 | 0.05 | 0.05 | 0.22 | 0.05 | 0.02 | 0.02 | 0.03 | 0.03 | 0.20 | 0.03 | 0.32 | 0.02 | 0.02 | 0.02 | 0.01 | 0.01 |
| Median | 18.87 | 1.12 | 3.31 | 0.83 | 0.14 | −0.30 | 18.70 | 5.97 | 0.19 | −0.93 | 0.86 | −0.39 | 0.60 | −0.60 | 0.18 | −0.27 | 0.24 | −0.44 | 0.49 | −0.28 | 3.15 | 0.08 | 0.03 | 0.00 | 0.02 | −0.01 |
| SD | 6.71 | 0.41 | 1.77 | 0.51 | 0.11 | 0.05 | 4.60 | 2.01 | 0.21 | 0.01 | 0.22 | 0.24 | 1.01 | 0.25 | 0.11 | 0.08 | 0.15 | 0.15 | 0.92 | 0.14 | 1.52 | 0.09 | 0.10 | 0.09 | 0.06 | 0.02 |
| Coeffic.Var (%) | 35.51 | 38.68 | 48.76 | 66.23 | 73.33 | −17.24 | 23.83 | 30.41 | 77.78 | −1.11 | 26.51 | −64.86 | 116.09 | −37.31 | 61.11 | 27.59 | 53.57 | −31.25 | 115.0 | −56.0 | 44.19 | 90.0 | 125 | −450 | 100 | −100.00 |
| S.Variance | 45.07 | 0.17 | 3.13 | 0.26 | 0.01 | 0.01 | 21.17 | 4.05 | 0.04 | 0.01 | 0.05 | 0.06 | 1.03 | 0.06 | 0.01 | 0.01 | 0.02 | 0.02 | 0.85 | 0.02 | 2.30 | 0.01 | 0.01 | 0.01 | 0.01 | 0.00 |
| Kurtosis | −0.92 | −1.14 | −0.79 | −0.49 | 1.94 | 0.19 | 1.22 | 1.67 | 7.83 | 1.53 | −0.10 | 0.30 | 2.90 | 8.42 | −1.10 | −0.98 | 6.37 | 2.53 | 13.56 | −0.23 | 1.13 | 1.24 | 2.41 | 21.4 | 0.20 | 5.35 |
| Skewness | 0.08 | 0.12 | 0.31 | 0.14 | 1.35 | −0.37 | 1.11 | 1.59 | 2.71 | 0.39 | −0.34 | −0.05 | 1.78 | −2.45 | −0.03 | −0.36 | 2.17 | −1.32 | 3.39 | 0.87 | 0.94 | 1.40 | 1.79 | −4.60 | 1.16 | −2.19 |
| Range | 24.29 | 1.36 | 6.33 | 1.91 | 0.43 | 0.21 | 18.45 | 7.15 | 0.89 | 0.45 | 0.76 | 0.98 | 3.67 | 1.20 | 0.35 | 0.29 | 0.68 | 0.64 | 4.49 | 0.49 | 6.37 | 0.34 | 0.37 | 0.46 | 0.22 | 0.09 |
| Minimum | 6.88 | 0.42 | 1.11 | −0.02 | 0.01 | −0.40 | 13.20 | 4.73 | 0.12 | −1.13 | 0.41 | −0.91 | 0.01 | −1.59 | 0.01 | −0.46 | 0.13 | −0.92 | 0.05 | −0.43 | 1.20 | 0.01 | 0.01 | −0.43 | 0.00 | −0.09 |
| Maximum | 31.17 | 1.78 | 7.45 | 1.89 | 0.44 | −0.19 | 31.65 | 11.88 | 1.01 | −0.68 | 1.17 | 0.07 | 3.71 | −0.39 | 0.37 | −0.17 | 0.81 | −0.28 | 4.53 | 0.07 | 7.57 | 0.35 | 0.37 | 0.03 | 0.22 | 0.00 |
| Sum | 404.46 | 23.34 | 79..97 | 16.94 | 3.26 | −6.38 | 423.84 | 145.33 | 5.85 | −19.76 | 18.34 | −8.18 | 19.13 | −14.74 | 3.86 | −6.43 | 6.19 | −10.54 | 17.69 | −5.48 | 75.71 | 2.29 | 1.73 | −0.54 | 1.24 | −0.36 |
| Obs. | | | | | | | | | | | | | | 22 | | | | | | | | | | | | |
| Jarque-Bera | 0.79 | 1.24 | 0.91 | 0.28 | 10.17 | 0.54 | 5.90 | 11.86 | 83.19 | 2.72 | 1.33 | 0.09 | 19.37 | 87.33 | 1.12 | 1.36 | 54.37 | 12.27 | 210.5 | 2.80 | 4.42 | 8.58 | 17.08 | 498 | 4.99 | 43.79 |
| p-value | 0.67 | 0.54 | 0.63 | 0.87 | 0.01 | 0.76 | 0.05 | 0.00 | 0.00 | 0.26 | 0.51 | 0.95 | 0.00 | 0.00 | 0.57 | 0.50 | 0.00 | 0.00 | 0.00 | 0.25 | 0.11 | 0.01 | 0.00 | 0.00 | 0.08 | 0.00 |

| | Explanatory variables | | | |
| | LAND | LABOUR | GDPpc | INF |
|---|---|---|---|---|
| Mean | 1.20 | 1273.33 | 4321.04 | 5.22 |
| Standard Error | 0.03 | 983.02 | 266.33 | 0.43 |
| Median | 1.22 | 280.28 | 4476.20 | 4.98 |
| SD | 0.16 | 4610.79 | 1249.20 | 2.02 |
| Coeffic.Var (%) | 13.33 | 362.10 | 28.91 | 38.70 |
| Variance | 0.03 | 21,259,386.02 | 1,560,497.36 | 4.07 |
| Kurtosis | −0.96 | 22.00 | −0.32 | −0.25 |
| Skewness | −0.23 | 4.69 | −0.79 | 0.40 |
| Range | 0.50 | 21,669.48 | 4133.41 | 7.24 |
| Minimum | 0.94 | 246.74 | 1808.88 | 2.21 |
| Maximum | 1.44 | 21,916.22 | 5942.29 | 9.45 |
| Sum | 26.44 | 28,013.30 | 95,062.79 | 114.83 |
| Observations | | | 22 | |
| Jarque-Bera | 1.04 | 524.20 | 2.37 | 0.65 |
| p-value | 0.60 | 0.00 | 0.31 | 0.72 |

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
