# Peer review of "Competitiveness of Namibia’s Agri-Food Commodities: Implications for Food Security"

_resources, doi:10.3390/resources12030034_

Round 1

Reviewer 1 Report

General Comment

This study examines the competitiveness of Namibia's agri-food sector and its implications for food security. The impact of global food commerce on food security is a pressing social and political issue throughout the world, notably in African countries. This study contributes to a better understanding of the determinants influencing agri-food commodity competitiveness in Namibia and, more broadly, in Sub-Saharan Africa.

The paper is well-structured. The introduction clearly defines the subject being analysed, and some important literature on the topic is examined. Some changes might be made to the way the results are presented to make them easier to understand.

Additional comments

1)      In Table 1 displays both the variables used to compute the indices and the variables employed in model estimation (from MEC forward). They should, in my opinion, be presented separately and better explained. It is particularly unclear how the variable HFIAS was calculated.

2)      It is not required to write the source of the tables if they were created by the authors.

3)      Lines 205-206: please clarify that the meaning of RCA>1 is right; I believe you mean higher rather than lower in line 205.

4)      In Equation 2  (line 233), the notation for exports for the sector “i” of country “j” is different from the one presented in Table1. I think that you should keep the same notation for imports and exports and that “N” should be replaced by “n”.

5)     Equation 4: while its meaning is clear, xi is not defined in Table 1.

6)     Equation 6: At first glance, it appears that the authors estimated a single model with RCA/LFI as the dependent variable; however, for clarity, I believe that the two models should have their own equation.

7)     Using brackets to display the data and results for the LFI model in Table 2 and following is not a good idea. Typically, brackets are used for standard deviation or t-values; I recommend including two columns for each item, one for RCA and one for LFI.

8)     In the Conclusions section, the policy recommendation in lines 548-551 should be better explained because governments typically struggle to reduce primary production exports while increasing value addition, thereby improving food security and reducing reliance on imports.

Author Response

Dear Reviewer,

Please find attached response to your comments.

With regards, 

Reviewer 2 Report

The articles investigate the factors that affect the agri-food comparative advantage of Namibia. The topic is relevant. The RQ is defined, the methodology applied is adequate, and the analysis is carried out correctly.

The limitation of the study should the added to the conclusions.

I suggest adding some other relevant references:

Balogh, J.M. and Jámbor, A. (2017), "The global competitiveness of European wine producers", British Food Journal, Vol. 119 No. 9, pp. 2076-2088. https://doi.org/10.1108/BFJ-12-2016-0609

Gois, T.C., Thomé, K.M. and Balogh, J.M. (2022), "Behind a cup of coffee: international market structure and competitiveness", Competitiveness Review, Ahead-of-print. https://doi.org/10.1108/CR-10-2021-0141

Balogh, J. M., & Jámbor, A. (2017). Determinants of revealed comparative advantages: The case of cheese trade in the European Union. In Acta Alimentaria (Vol. 46, Issue 3, pp. 305–311). https://doi.org/10.1556/066.2016.0012

Author Response

(The authors gave the same response as above.)

Reviewer 3 Report

The manuscript entitled "Competitiveness of Namibia's Agri-food Commodities: implications for food security" (resources-2182410) was aimed at assessing the degree of Namibia's agri-food products competitiveness in relation with its determinants in the light of food insecurity concerns.

To improve the quality of the manuscript, I propose the following:

(1) Please consider refining your abstract, according to the journal's recommendation, especially according to this point: the abstract should include information on the background – a broad and brief presentation of the general context of the analyzed issue. Line 17: It is not clear to what you are referring to when you used the word "these" – What are the agri-food commodities / the major export category specifically?

(2) The Introduction section should include a final paragraph dedicated to presenting the structure of this empirical study.

(3) Methodologically, I suggest further developing the reasoning for integrating alternatives (which I appreciated) to competitiveness indices (subsection 2.2). For reference, I also recommend this paper: Istudor, N., Constantin, M., Ignat, R., Chiripuci, B.-C., & Petrescu, I.-E. (2022). The Complexity of Agricultural Competitiveness: Going Beyond the Balassa Index. Journal of Competitiveness, 14(4), 61–77. https://doi.org/10.7441/joc.2022.04.04. Moreover, I agree that land and labour productivity are suitable variables for the assessment of competitiveness levels and I suggest grounding this approach in the light of  Porter's traditional Diamond Model. For reference, this is a paper that applied Porter's Diamond Model to the specific case of an agri-food chain: Constantin, M.; Sacală, M.-D.; Dinu, M.; PiÈ™talu, M.; Pătărlăgeanu, S.R.; Munteanu, I.-D. Vegetable Trade Flows and Chain Competitiveness Linkage Analysis Based on Spatial Panel Econometric Modelling and Porter’s Diamond Model. Agronomy 2022, 12, 411. https://doi.org/10.3390/agronomy12020411.

(4) HFIAS was introduced in the conceptual framework (Figure 1) before explaining how it is connected with the competitiveness assessment in a model (line 325). I suggest deciding if this is the proper approach of the methodological flow. Right now, it appears rather confusing.

(5) I suggest moving tables 2-6 to the appendix of the paper. Please consider adding more descriptive statistics in table 7 (standard deviation, coefficient of variation, Skewness, Kurtosis, Jarque-Bera) and interpret the results.

(6) I have a concern regarding the explanatory variables from table 7 – How were land and labour productivity integrated in a model specific to different classes of agri-food products? For example, was land productivity determined systematically for cereals, vegetables (and so on), and later integrated into the model, systematically?

(7) Line 458: Regression results for the HFIAS; line 471: Regression results for the HFIAS? Please revise.

(8) The discussion feels rather rushed, as now there are few lines (522-526) that compare the results of this empirical study to those of other authors that have published in the same field. Please further expand this part of the paper. Moreover, I was expecting a better connection between the analysis of competitiveness and food security.

(9) In the Conclusions section, when discussing the relation between the SDGs and competitiveness, I suggest briefly touching on the issue of the divergent actions required to achieve such sustainable objectives. For reference, I suggest: Constantin, M.; Sapena, J.; Apetrei, A.; Pătărlăgeanu, S.R. Deliver Smart, Not More! Building Economically Sustainable Competitiveness on the Ground of High Agri-Food Trade Specialization in the EU. Foods 2023, 12, 232. https://doi.org/10.3390/foods12020232. Moreover, in the Conclusions section, please approach the current limitations of this empirical study and mention some future research avenues. Approaching more managerial implications would also add more value to the paper.

Author Response

(The authors gave the same response as above.)

Round 2

Reviewer 3 Report

The authors have improved the quality of the manuscript according to the suggestions.